# Vaccinated Patients Admitted in ICU with Severe Pneumonia Due to SARS-CoV-2: A Multicenter Pilot Study

**DOI:** 10.3390/jpm11111086

**Published:** 2021-10-25

**Authors:** Ángel Estella, Mª Luisa Cantón, Laura Muñoz, Isabel Rodriguez Higueras, María Recuerda Núñez, Julia Tejero Aranguren, Benito Zaya, Carmen Gómez, Rosario Amaya, Ángela Hurtado Martinez, María del Valle Odero Bernal, Carmen De la Fuente, Juan Carlos Alados, Jose Garnacho-Montero

**Affiliations:** 1Intensive Care Unit, Medicine Department University of Cadiz, University Hospital of Jerez, INiBICA, 11001 Jerez, Spain; 2Intensive Care Unit, University Hospital Virgen Macarena, 41013 Sevilla, Spain; luisabulnes@hotmail.com (M.L.C.); jose.garnacho.sspa@juntadeandalucia.es (J.G.-M.); 3Intensive Care Unit, Hospital Costa del Sol de Marbella, 29001 Málaga, Spain; lauramunozmendez@gmail.com (L.M.); zayaganfo@gmail.com (B.Z.); c_gomezgonz@hotmail.com (C.G.); 4Intensive Care Unit, University Hospital Torrecárdenas, 04001 Almería, Spain; mrodriguezhigueras@yahoo.es; 5Intensive Care Unit, University Hospital of Jerez, INiBICA, 11001 Jerez, Spain; mariamongongo@hotmail.com (M.R.N.); valleob@gmail.com (M.d.V.O.B.); juanc.alados@gmail.com (J.C.A.); 6Intensive Care Unit, University Hospital San Cecilio, 18001 Granada, Spain; juliatejero@hotmail.com; 7Intensive Care Unit, University Hospital Virgen del Rocío, 41013 Sevilla, Spain; rosario.amaya.villar.sspa@juntadeandalucia.es; 8Intensive Care Unit, University Hospital Virgen de Valme, 41013 Sevilla, Spain; angela.hurtado.sspa@juntadeandalucia.es; 9Intensive Care Unit, University Hospital Reina Sofía, 14004 Córdoba, Spain; carmen.fuente.sspa@juntadeandalucia.es

**Keywords:** SARS-CoV-2, severe pneumonia, acute respiratory distress syndrome, vaccine, ICU, Delta variant SARS-CoV-2

## Abstract

**Background:** The aim of this study was to analyze the percentage of patients admitted to the ICU having received the vaccine against COVID-19, to describe the clinical profile of vaccinated patients admitted to the ICU, and to assess the humoral immune response to vaccination. **Methods:** In this multicenter prospective descriptive cohort study, consecutive critically ill patients with confirmed SARS-CoV-2 pneumonia who received at least one dose of the SARS-CoV-2 vaccine were included. The time of study was from 1 July to 10 August of 2021. **Results:** Of the 94 consecutive patients from seven Andalusian ICUs admitted during the time of study, 50 (53.2%) received at least one dose of anti SARS-CoV-2 vaccine. No patient was admitted having previously had SARS-CoV-2 infection. The B.1.617.2 (Delta) variant was the most frequently identified, in 80.76% of cases. Patients with a complete vaccination with non-optimal antibody levels were immunocompromised. Fifteen patients were admitted to the ICU with Acute Respiratory Distress Syndrome (ARDS) without having completed their vaccination; the clinical profile was younger and with less comorbidities compared to patients with full vaccination. There were no differences in severity of ARDS. **Conclusions:** Most of the patients who were admitted to the ICU having received a dose of the vaccine were not optimally vaccinated; fully vaccinated patients who did not obtain optimal serum antibody levels were patients considered immunocompromised.

## 1. Introduction

Although there has been a decrease in the number of hospitalizations when compared to the previous waves of the pandemic [1], severe cases continue to be admitted to ICUs, most of them without yet having access to vaccination. The hope generated by vaccines against SARS-CoV-2 has been satisfied with the observation of Intensive Care Units (ICUs) of a decrease in the number of patients admitted. While current vaccines are all highly effective at preventing severe cases of SARS-CoV-2 no vaccine is 100% effective, which means breakthrough infections in vaccinated patients can occur [2].

On the date on which the recruitment of patients from this study was closed, a total of 10,763,537 doses of vaccines had been administered in the Autonomous Community of Andalusia, at a vaccination rate similar to the rest of the Spain and similar to that of the rest of European countries. This is assuming, then, approximately 60% of the adult population [3].

There are no previous studies published about vaccinated patients with severe breakthrough pneumonia. It seems important to know whether vaccine effectiveness against severe COVID-19 results in ICU admission. With this pilot study we can estimate the future burden of this disease in autumn and learn more about what type of patients will be admitted to ICUs infected by SARS-CoV-2 while vaccinated. We hypothesized that vaccinated patients who are developing severe forms of the disease are due to incorrect vaccination schedule or immune status-related comorbidities that have conditioned an insufficient immune response. We present this pilot study with the following objectives: to analyze the percentage of patients admitted to the ICU having received the vaccine against COVID-19, to describe the clinical profile of vaccinated patients admitted to the ICU for severe pneumonia due to SARS-CoV-2, and to assess the humoral immune response to vaccination.

## 2. Methods

Design: Multicenter prospective descriptive cohort study.

Setting: Seven intensive care units (ICU) participated in the study from Andalusian region in Spain.

Patients or participants: Consecutive critically ill patients with confirmed SARS-CoV-2 infection admitted to the participant ICUs from July of 2021 to August 10th were included. Inclusion criteria were patients older than 18 years admitted to the ICU due to severe pneumonia by SARS-CoV-2confirmed by RT-PCR, who received at least one dose of SARS-CoV-2 vaccine. Exclusion criteria were unvaccinated patients with SARS-CoV-2 pneumonia, SARS-CoV-2 infection but admitted to ICU for other reason not related to COVID-19 pneumonia, and patients or relatives who declined to participate in the study.

Variables: We analyzed demographic characteristics, comorbidities, type of vaccine and number of doses, time from vaccine administration to ICU admission, severity score by APACHE II and organ failure by SOFA score, and respiratory function at ICU admission and need for support therapy in ICU.

Interventions: To evaluate an individual’s humoral response to vaccines, levels of antibodies were measured. SARS-CoV-2 IgG II Quant assay was performed on the Abbott Alinity i platform in accordance with the manufacturer’s package insert [4], this assay is designed to detect immunoglobulin class G (IgG) antibodies to the receptor binding domain (RBD) of the S1 subunit of the spike protein of SARS-CoV-2 in serum and plasma.

Due to the nature and contact restrictions of this pandemic disease, the informed consent was obtained from the patient when it was possible or from the relatives by a phone call with or without a complementary mail, and it was registered in the medical history. A sub-study within the Andalusian SAMIUCOVID registry evaluated by the Ethical Committee of Investigation in Cádiz [SAM-COVUCI-2020-01] approved the study protocol.

Immunocompromised patients were defined as patients with malignancy, transplantation status, or receipt of immunosuppressive treatment. A breakthrough infection was defined as the detection of SARS-CoV-2 on RT-PCR assay performed at hospital admission in vaccinated patients with clinical suspicion of pneumonia with bilateral infiltrate and hypoxemia.

For the analysis of the groups, we differentiated whether they had had a previous SARS-CoV-2 infection. We distinguished one group called completed vaccination and another group as incomplete vaccination based on not having received the total doses and recommended time after the last dose. Depending on the quantification of antibodies we distinguished as optimal or suboptimal antibody level.

Statistical analysis: A descriptive analysis was made using absolute frequencies for qualitative variables and mean with standard deviation (SD) or median with percentiles 25th and 75th (pp. 25–75) when appropriate.

## 3. Results

A total of 94 consecutive patients from seven Andalusian ICUs were admitted during time of study, and 50 received at least one dose of anti SARS-CoV-2 vaccine. None of the patients had previously been infected with SARS-CoV-2, 35 had received the full vaccination schedule and 15 had not received their second dose and therefore their vaccination status was considered incomplete. Figure 1 shows flowchart of patients.

This number of admissions accounts for 53.2% of enrolled ICU admissions during time of study. Of the studied patients, 32% received Pfizer and BioNTech’s COVID-19 vaccine, 30% the Oxford/AstraZeneca COVID-19 vaccine, 22% Moderna’s COVID-19 vaccine and 16% Janssen Pharmaceutical vaccine. Table 1 shows clinical characteristics of vaccinated patients detailing clinical features, comorbidities, supportive treatments, and severity.

No patient was admitted having previously had SARS-CoV-2 infection. In 26 patients, in whom the identification of the virus strain was available, the B.1.617.2 (Delta) variant was the most frequently identified at 80.76%, the B117UK variant accounted for 15.38% of patients and the B.1.351 variant 3.84%. According to the recommendations of the health authorities, 35 patients, 70%, received the full vaccination schedule. We can differentiate between different clinical profiles in patients depending on whether or not they have received their full vaccination, as shown Table 2. There were not statistical differences in variables analyzed.

Another important factor was the number of days between receiving the last dose of vaccine until ICU admission; 11 completely vaccinated patients were admitted to the ICU between the first 21 days after vaccine administration, 9 of 11, or 81.8%, had adequate levels of antibodies. One of the patients of this group with suboptimal level of antibodies was an immunocompromised patient.

Of the remaining patients with a complete vaccination received beyond three weeks after the last dose of vaccine, four patients did not have optimal antibodies levels and all of them were immunocompromised.

Fifteen patients were admitted to the ICU with Acute Respiratory Distress Syndrome (ARDS) without having completed their vaccination, their clinical profile is shown in Table 2. In the 10 patients in whom antibody levels could be determined at admission to the ICU, the total number of patients did not reach protective levels. Seven patients were admitted to the ICU within the first 21 days after vaccine administration and one was immunocompromised.

## 4. Discussion

SARS-CoV-2 vaccination is one of the greatest challenges in modern public health. The main achievement in the face of this global challenge has been the decrease in the number of hospitalizations [5]; the role of vaccines preventing new severe acute respiratory syndrome coronavirus 2 (SARS-CoV-2) infections has been scarcely studied. The fifth wave of the pandemic in Spain is marked by the start of implementation of mass vaccination against SARS-CoV-2. No patient admitted to the ICU had previous SARS-CoV-2 infection; these people have been shown to have a better serological response to the administration of vaccines [6].

This pilot study aims to investigate the role of vaccination in severe cases and to explain the effectiveness of the vaccines, especially of vaccinated people admitted to the ICU. The main finding that we can deduct from this analysis is that most of the patients who have been admitted to the ICU having received a dose of the vaccine were not optimally vaccinated, understood as such to have received the recommended doses and wait 21 days after the administration of the last dose [7,8]. Recently, it has been documented that effectiveness after one dose of vaccine was notably lower among persons with the Delta variant which was predominant in our patient series [9], it seems less likely widespread escape from responses generated by vaccination with variant B.1.1.7 [10].

Only half of the patients recruited had met the appropriate doses and vaccination deadlines when they were admitted to the ICU, and of these, those who did not obtain optimal serum antibody levels were patients considered immunocompromised [11,12]. This reinforces the need to strengthen all other preventive measures since this pandemic cannot be overcome at this stage when herd immunity has not been achieved only with vaccination [13]. According to our findings, vaccination protocols need to be improved and protective measures against the virus in immunocompromised patients need to be strengthened; in the study, this group of patients were serologically non-responders. If with personalized medicine we are learning that treatments cannot be universalized with the administration of vaccines something similar happens, the clinical profile and comorbidities make it necessary to start proposing a strategy based on personalized medicine for vaccination, being demonstrated that there will be patients who with the standardized vaccine not enough protection is achieved.

Future studies are necessary to assess if administration of additional vaccine doses may be a potential strategy for these patients who were excluded from all vaccine trials [14]. In addition, the predominance of the role of neutralizing antibodies versus the cellular response raises us to consider that fully vaccinated patients with negative serology should receive a third dose of vaccine.

These findings lead us to think with hope that when the population is fully vaccinated, the incidence of severe SARS-CoV-2 pneumonia will drop considerably. The clinical profile of patients was different according to the degree of vaccination, and the established vaccination schedule in which the age groups of the elderly were the first to access the vaccine conditioned our findings in which we observed differences in the profile of patients admitted to the ICU. Those who had not completed their vaccination were younger and with less comorbidity, and an interesting finding was that there were no differences in the level of severity or organ failure. Severity of ARDS was similar to pre-vaccination time [15,16]. The present study was designed to seek explanation for the fact that vaccinated patients are being admitted to the ICU. It is a multicenter pilot study in which we were able to analyze what occurs both in fully vaccinated patients and those who have not yet completed vaccination. Being fully vaccinated seems to protect against the development of severe forms of the disease except in immunocompromised patients.

The main limitations of the study are that we have not followed up on patients, most patients are still admitted to the ICU, and the short recruitment time of the study. Determination of antibody levels is not part of a routine procedure in hospitalized patients and therefore was not available in a small part of the sample, and we have also been able to observe that having adequate levels does not always ensure not developing the disease, considering the importance that other immunological mechanisms such as cellular immunity occupy in the pathophysiology of this disease [17]. We observed that the first dose of vaccination does not prevent the development of severe forms of the disease, supporting to maximize vaccine uptake with two doses among vulnerable populations.

The most novel aspect of the study is that we present the first series of critically ill patients, most of them with invasive mechanical ventilation requirements, who have received vaccination against SARS-CoV-2. In the coming months when the population’s mass vaccination strategies have ended the predominant clinical profile of ICU admissions will likely be of elderly patients with comorbidity and immunocompromised patients.

The next step in future studies will be to know the clinical evolution, and if there are differences in the prognosis of vaccinated patients who have been admitted to the ICU. In addition, investigating what factors predispose to the development of serious cases despite vaccination will open a door to the future of research based on personalized medicine [18].

## Figures and Tables

**Figure 1 jpm-11-01086-f001:**
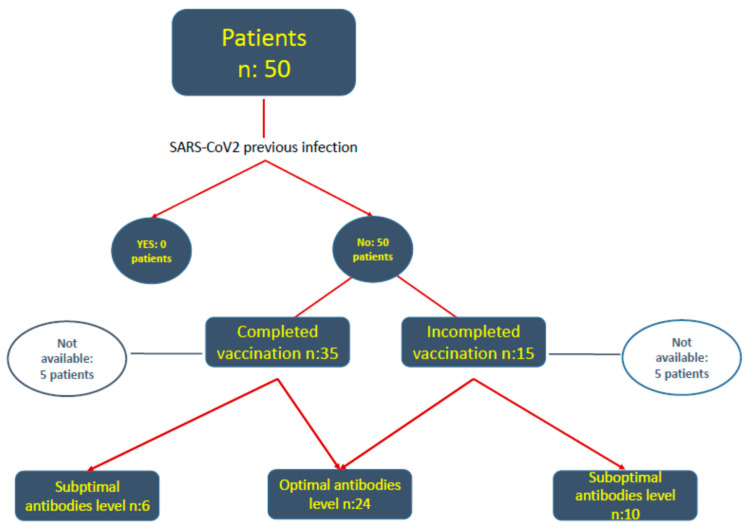
Shows flowchart of patients.

**Table 1 jpm-11-01086-t001:** Baseline characteristics of patients included in the study.

Variables	Patients (*n* = 50)
**Demographic Characteristics**	
Age (years)	61.5 (24–84)
Gender	
Male	41 (82%)
Female	9 (18%)
BMI (Kg/m^2^)	29.34 (22–56.6)
**Comorbidities**	
Any comorbidity	40 (80%)
Hypertension	24 (48%)
Heart disease	12 (24%)
Respiratory disease	15 (30%)
COPD	3 (6%)
Chronic kidney disease	8 (16%)
Immunosuppression	40 (3·6%)
Hematological disease	7 (14%)
Autoimmune disease	12 (24%)
Neurologic disease	3 (6%)
Cirrhosis	4 (8%)
**Respiratory Support Treatment**	
Non-invasive ventilation	7 (14%)
High-flow oxygen therapy	43 (86%)
Invasive ventilation	37 (74%)
**Disease Severity**	
APACHE II score	10.5 (4–31)
SOFA score	4 (2–16)
ARDS	
Mild ^1^	0%
Moderate ^2^	24.5%
Severe ^3^	75.5%

Data are presented as numbers (%) or medians (interquartile range). ^1^ Defined as the worst value of PaO_2_/FiO_2_ ratio < 300 within the first day of ICU admission. ^2^ Defined as the worst value of PaO_2_/FiO_2_ between 200–300 within the first day of ICU admission. ^3^ Defined as the worst value of PaO_2_/FiO_2_ < 100 within the first day of ICU admission. BMI: Body mass index. COPD: Chronic obstructive pulmonary disease. APACHE: Acute Physiology And Chronic Health Evaluation. SOFA: Sequential Organ Failure Assessment. ARDS: Acute respiratory distress syndrome.

**Table 2 jpm-11-01086-t002:** Clinical profile of patients according to grade of vaccination.

Variables	Fully Vaccinated Patients (*n* = 35)	Partially Vaccinated Patients (*n* = 15)
**Demographic Characteristics**		
Age (years)	64 (44–84)	57.5 (24–73)
Gender		
Male	28 (80%)	13 (86.6%)
Female	7 (20%)	2 (13.3%)
BMI (Kg/m^2^)	28.68 (22.04–43.25)	30.91 (23.6–46.3)
**Comorbidities**		
Any comorbidity	33 (94.28%)	12 (80%)
Hypertension	21 (60%)	6 (40%)
Heart disease	9 (25.7%)	5 (33.3%)
Respiratory disease	11 (31.4%)	2 (13.3%)
COPD	3 (8.6%)	0 (0%)
Chronic kidney disease	7 (20%)	4 (26.6%)
Immunosuppression	10 (28.57%)	0 (0%)
Hematological disease	6 (17.1%)	1 (6.6%)
Neurologic disease	2 (5.7%)	1 (6.6%)
Cirrhosis	3 (8.6%)	2 (13.3%)
**Respiratory Support Treatment**		
Non-invasive ventilation	7 (20%)	0 (0%)
High-flow oxygen therapy	30 (85.7%)	15 (100%)
Invasive ventilation	23 (65.7%)	12 (80%)
**Disease Severity**		
APACHE II score	12 (4–31)	11 (5–27)
SOFA score	3 (2–16)	5.5 (3–12)
ARDS		
Mild ^1^	0%	0%
Moderate ^2^	26.5%	26.6%
Severe ^3^	73.5%	73.3%

Data are presented as numbers (%) or medians (interquartile range). ^1^ Defined as the worst value of PaO_2_/FiO_2_ ratio < 300 within the first day of ICU admission. ^2^ Defined as the worst value of PaO_2_/FiO_2_ between 200–300 within the first day of ICU admission. ^3^ Defined as the worst value of PaO_2_/FiO_2_ < 100 within the first day of ICU admission. BMI: Body mass index. COPD: Chronic obstructive pulmonary disease. APACHE: Acute Physiology And Chronic Health Evaluation. SOFA: Sequential Organ Failure Assessment. ARDS: Acute respiratory distress syndrome.

## Data Availability

Data and material of the study is available for editorial team and reviewer of the journal.

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
