# Peer review of "Vaccinated Patients Admitted in ICU with Severe Pneumonia Due to SARS-CoV-2: A Multicenter Pilot Study"

_jpm, 2021, doi:10.3390/jpm11111086_

Round 1

Reviewer 1 Report

The authors studied 94 COVID-19 vaccinated patients by analyzing the SARS-CoV-2 IgG response. They found that 80.76% of patients admitted to ICU are infected with B.1.617.2(delta). 15 out of 94 patients were admitted with ARDS.

Please consider rewriting the paper with improvement in the flow chart (barely visible). Please expand the sections especially, the Introduction. Building a hypothesis would help readers to get the data coherently.  Please consider mentioning the Table with descriptive legends.

Author Response

Dear reviewer, thank you for your interest in the manuscript and considerations that will certainly help improve the quality of the manuscript. Here's a point-by-point response to your comments:

 -. According to your suggestion I have improved the quality of Figure 1, 

     republishing it in a more refined format that improves its visibility.

 -.  Following your instructions we have expanded the introduction of the

     manuscript attending to your request we have established a study     

     hypothesis to help the reader to a friendlier reading.

 -. According to your request, we mention the tables with descriptive legends.

Thank you for your recommendations.

Reviewer 2 Report

Thank you very much for giving me the opportunity to the review this manuscrpt focuses on analyzing the effects of vaccination in patients admited in ICU with severe pneumonia due to SARSCoV2. Authors have elaborated a manuscript about a topic of interest for readers and scientific community. However, manuscript should be presented in a structured way following the criteria of high scientific quality.

Firstly, introduction is poorly developed . Authors should provide a rational background of the SARSCoV2, effects and impact of vaccination in the population. Also, it is recommended that authors provide information about the rates of vaccination in Spain in particular as comparing with other contries, and how vaccination may impact in the number of infected patients and severity of illness. Then, explain why this article is important.

In the method section, authors should clarify different sections including: participants, measures/instruments/variables and procedure. Also, statistical analyses should be included in a separated section. In the method section, how antibodies were measured?

In the results section, figure 1 should be improved, number of participants and their percentages should be included into text. In table 2, authors should include the differences obtained between both groups (full and partially vaccinated) and then control for confusors (maybe age, sex..). It is also recommended to included three sub-sections according to the aims (percentages of vaccination and infections; time; inmune response.

In the discusión, authors should explain how clinicians should take into account the present findings? And how results fit with other current studies. In addition, how the present results provide a contribution to personalized medicine.

Author Response

Dear reviewer, thank you for your interest in the manuscript and considerations that will certainly help improve the quality of the manuscript. I agree with your revision and I believe that by following your successful recommendations we can improve the content of the manuscript

Here's a point-by-point response to your comments:

 -. Following your instructions we have expanded the introduction. We provide in the new versión information about rates of vaccination in Spain and comparing with other countries. We have established a study hypothesis about impact in ICU admission and severity of illness.

  -. According with your recommendations methods section was modified and improved.

  -. In the results section, figure 1 has been improved, number of participants and their percentages has been included into text.

   -. Information about table 2 has been modified in the text according your recommendations. There were not statistical differences in clinical features and comorbidities.

   -. Discussion has been improved with the inclusión of several novel aspects suggested by you. We have included a paragraph about personaliced medicine in vaccination according with your recommendations.

Round 2

Reviewer 2 Report

Thank you to the authors for this improved version of the manuscript. Authors have made substantial changes. However, I suggest some concerns that need to be improved. Table 1 is overlaping with text. Also, information is a little bit confused due to overlaping.